# Polyurethane-Carbon Nanotubes Composite Dual Band Antenna for Wearable Applications

**DOI:** 10.3390/polym12112759

**Published:** 2020-11-23

**Authors:** Robert Olejník, Stanislav Goňa, Petr Slobodian, Jiří Matyáš, Robert Moučka, Romana Daňová

**Affiliations:** 1Centre of Polymer Systems, University Institute, Tomas Bata University in Zlín, 76001 Zlín, Czech Republic; slobodian@utb.cz (P.S.); matyas@utb.cz (J.M.); moucka@utb.cz (R.M.); danova@utb.cz (R.D.); 2Faculty of Applied Informatics, Tomas Bata University in Zlín, 76005 Zlín, Czech Republic; gona@utb.cz

**Keywords:** polymer antenna, composite material, carbon nanotubes, effective permittivity, effective permeability, effective conductivity, wearable electronics

## Abstract

The design of a unipole and a dual band F-shaped antenna was conducted to find the best parameters of prepared antenna. Antenna radiator part is fully made of polymer and nonmetal base composite. Thermoplastic polyurethane (PU) was chosen as a matrix and multi-wall carbon nanotubes (MWCNT) as an electrical conductive filler, which creates conductive network. The use of the composite for the antenna has the advantage in simple preparation through dip coating technique. Minor disadvantage is the usage of solvent for composite preparation. Composite structure was used for radiator part of antenna. The antenna operates in 2.45 and 5.18 GHz frequency bands. DC conductivity of our PU/MWCNT composite is about 160 S/m. With this material, a unipole and a dual band F antenna were realized on 2 mm thick polypropylene substrate. Both antenna designs were also simulated using finite integration technique in the frequency domain (FI-FD). Measurements and full wave simulations of S_11_ of the antenna showed good agreement between measurements and simulations. Except for S_11_, the gain and radiation pattern of the antennas were measured and simulated. Maximum gain of the designed unipole antenna is around −10.0 and −5.5 dBi for 2.45 and 5.18 GHz frequency bands, respectively. The manufactured antennas are intended for application in wearable electronics, which can be used to monitor various activities such as walking, sleeping, heart rate or food consumption.

## 1. Introduction

The rise of the Internet of Things (IoT) in the 21st century has seen a pressing need of developing easily incorporable electrochemical devices for acquiring real-time data. One segment of this field comprises smart clothing and wearable electronics, which is used for diverse ends ranging from activity monitoring bracelets, smart watches or glasses over GPS enabled shoes to life-saving devices used in healthcare [1,2]. IoT devices generally consist of a sensing unit collecting vital data and a transmitting unit, which sends the collected data to the processing or display unit. These are typically rigid, conventional electronic parts in the flexible plastic or elastic rubber substrates which presents some limitations in their seamless integration e.g., into clothing. Thus, developing and designing miniature and/or highly flexible IoT devices is highly desirable and currently draws a great deal of interest. Researchers have thus focused on developing wearable formats of electrochemical devices since they can play a vital role in the field of personalized IoT [3,4,5,6].

Antenna design which could be used in the segment of wearable electronics often features antennas with non-metallic radiators. If the radiator is to be made of a non-metallic material then this still has to have sufficiently high conductivity at microwave frequencies to achieve sufficient radiation efficiency and consequently the highest possible gain. Therefore, conductive textiles, conductive polymers [7] or composite materials are mostly used. Conductive textiles are employed in fabric antennas, which are most frequently in the form of a patch antenna (a rectangular microstrip antenna). These antennas of good radiation characteristics typically consist of upper and lower conductive layers of antenna patch while the ground plane and the middle layer are made of dielectric substrate [8,9].

Another large group of materials used in antenna with non-metallic radiators is conductive polymers. First attempts in the field were made with polyaniline (PANI) composites [10]. Later researchers moved on to the usage of carbon nanotubes [11,12] or other very good conductors (silver and gold particles) to increase electrical conductivity [13,14]. The use of carbon nanotubes has led to technologies using PANI and carbon fibers (PANI-MWCNT) [15,16], technologies using polypyrrole (PPy), [17] and PPy combined with PEDOT materials [18,19]. Technologies using pure PANI exhibit DC conductivity around 4500 S/m [15,16]. Polypyrrole materials have conductivity about 2000 S/m [17] and are typically around 100 μm thick even though stacking of PPy layers into final thickness of several hundreds of microns is also possible. PEDOT materials have higher conductivity of about 10,000 S/m but their thickness is limited approximately to 10 μm. Small physical thickness and conductivity around 10,000 S/m results in DC sheet resistance of PEDOT materials being larger than the sheet resistance of PPy material [17]. All above-mentioned materials can be used in the design of an antenna having large radiation efficiency. PANI-MWCNT showed radiation efficiency around 60% [15]. PPy solution also shows radiation efficiency around 60% [17]. Antennas with PEDOT material have typically lower radiation efficiency of around 30% [17]. Recently, other approaches in preparation of conductive polymers with polyaniline have appeared in the literature. These approaches are represented by multifunction poly/amide-imid and polyaniline films [20] and highly conductive poly/amide-imid films [21]. 

Apart from conductive polymers, other non-metal materials can be used in radiating part of microwave antennas with large radiation efficiency. Most common case is the use of composites with long carbon fibers [22] or combination of long carbon fibers with carbon nanotubes [22]. Another solution is the use of pure nanotubes [23]. The approach described in [23] leads to a material with conductivity as high as 50,000 S/m, also known as buckypaper [23]. Conductivity measurement of carbon nanotubes at microwave frequencies was carried out in detail in [24]; conductivity of pure carbon nanotubes is around 160,000 S/m and conductivity of buckypaper is around 50,000 S/m [25]. Through mixing of carbon nanotubes with a dielectric matrix, the effective conductivity of the composite becomes lower than the conductivity of pure nanotubes and strongly depends on its concentration and spatial distribution of carbon nanotubes within the composite [7]. Examples of measurement of the conductivity of composite materials in waveguides can be found in [26,27].

In this paper, a novel approach to preparation of conductive composite material (using a “dip” technique) suitable for antenna’s radiator is presented together with an actual antennas design. The novel approach uses a composite comprising polyurethane mixed with multiwall carbon fibers (PU/MWCNT). The advantage of this approach lies in the simplicity of material preparation, which is comparable to polyaniline and polypyrrole, but is significantly less demanding than for PEDOT. As both thickness and conductivity of the PU/MWCNT composite is limited antenna radiators must be made from relatively thick layers (hundreds of microns), to achieve substantial radiation efficiency of antennas. Required thickness is realized by stacking several layers on top of each other. Although from an antenna construction viewpoint, it would be more appropriate to design a dual band antenna of other topology (e.g., a microstrip patch, a slot, or a variant of PIFA antenna), for the sake of simplicity, we decided to study a simple unipole and F-antenna over ground plane. Even though the gain of this approach is smaller, it is still acceptable for short range operating devices; typically, our antennas show gain of −10.0 and −5.5 dBi at 2.45 and 5.18 GHz frequency band, respectively. The resulting antenna is lightweight and small enough to fit into a pocket and although it is not fully flexible, it can tolerate certain bending; thus could be placed on upper arms or on a thigh, where minimum bending occurs.

## 2. Materials and Methods

### 2.1. Materials

All the chemicals used were of analytical grade. *N*,*N*–dimethylformamide 99.5%, analytical reagent grade was purchased from Fisher Chemical (Waltham, MA, USA). Purified multi wall carbon nanotubes (MWCNT) were produced by the chemical vapor deposition (CVD) of acetylene supplied by Sun Nanotech Co. Ltd., Nanchang, Jiangxi, China, China. According to the supplier, the nanotubes have diameters of 10–30 nm, length 1–10 µm, purity >90% and electrical resistivity 0.12 Ω cm.

A Thermoplastic polyurethane (PU) Desmopan DP 385S was purchased from Bayer MaterialScience, (Leverkusen, Berlin, Germany). According to supplier’s specifications PU has following characteristic: strength of 48.9 MPa, with the strain at break of 442% and density of 1.20 g cm^−3^. Polyurethanes (PU) Desmopan^®^ is a thermoplastic block copolymer characterized by a wide range of properties. Its linear polymeric chains consist of alternating flexible, elastic segments. Polyurethane was used as a matrix and it was filled with MWCNT. It serves as an elastic base for the MWCNT antenna radiator layer.

### 2.2. Sample Preparation

Polyurethane solution 10 wt % in dimethylformamide (DMF) was prepared. The solution was mixed over night at 400 rpm and 90 °C. Subsequently carbon nanotubes were added to the solution and the dispersion was mixed using UP 400S ultrasonic homogenizer for 15 min, at power of 50% and at pulse of 50%. Then the dispersion was mixed mechanically using a magnetic stirrer for 30 min at 400 rpm. After that 30 wt %. composite was made in the form of dispersion. The deep coating method was used for this layer preparation. PET foil was used as a substrate. After drying the desired shape of antenna (Figure 1) was cut out.

### 2.3. Methods

The scanning electron microscopy (SEM) observations were made using FEI Nova NanoSEM, scanning microscope (Waltham, MA USA). Sample was placed on SPI double side carbon tape substrates. All specimens were sputtered with gold/Pb before imaging to improve conductivity. After cryo-fractured in liquid nitrogen, the morphology of the failure surfaces (cross section) of PU/MWNT sample was observed.

The transmission electron microscopy (TEM) observations were made using TEM, JEOL Ltd., Tokyo, Japan microscope. The dispersions of the MWCNTs in acetone were cast on Cu grids.

The materials were also analyzed by X-ray photoelectron spectroscopy (XPS) on TFA XPS Physical Electronics instrument (Chanhassen, MN, USA) [28,29] at the base pressure in the chamber of about 6 × 10^−8^ Pa. The samples were excited with X-rays over a 400 µm spot area with a monochromatic Al Kα1,2 radiations at 1486.6 eV. Photoelectrons were detected with a hemispherical analyzer positioned at an angle of 45° with respect to the normal to the sample surface. Survey-scan spectra were made at a pass energy of 187.85 eV, the energy step was 0.4 eV. Individual high-resolution spectra for C 1s were taken at a pass energy of 23.5 and 0.1 eV energy step. The concentration of elements was determined from survey spectra by MultiPak v7.3.1 software from Physical Electronics (Chanhassen, MN, USA).

Thermogravimetrical (TGA) studies were performed on a TGA Q500 (TA instruments, New Castle, DE, USA). The analysis was made under following conditions: temperate range from 25 to 1000 °C at 10 °C/min heating rate under nitrogen flow of 50 mL/min.

The conductivity of our composite material was measured on a 100 mm long and 10 mm wide sample made from polyethylene terephthalate (PET) substrate covered with a layer of PU/MWCNT composite. The composite and PET layers were 70 and 147 μm thick, respectively. DC resistance was measured in two-point set-up using Fluke 867B graphical multimeter (Eindhoven, The Netherlands); with probes located at the ends of the sample strip—good reproducibility of conductivity measurement was achieved.

Measurement of material properties of composite materials is typically done with the use of rectangular waveguides [26,27]. To measure the effective conductivity and the effective permittivity of the studied PU/MWCNT composite firstly the samples were placed inside the rectangular waveguides. These measurements relied on our previous work described in [30]. In the second measurement set-up, the samples were placed between the flanges of the rectangular waveguides. Finally, the third measurement of the effective conductivity were conducted on a microstrip test circuit.

For the design of the dual band antenna, two topologies were chosen. The first one was in the form of a simple unipole while the second in the form of an F-shaped antenna [31]. Dimensions of both antennas’ designs are shown in Figure 1. Basic measurements of the antennas were performed in the anechoic chamber using the EMC 32 measurement software from Rohde Schwarz (Munich, Germany). Measurements of reflection coefficient *S*_11_ of antennas was done with the aid of the Keysight handheld spectrum analyzer N9912A (Santa Rosa, CA, USA) and with the Agilent vector network analyzer PNA-L-N5230A (Santa Rosa, CA, USA). The *S*_11_ measurements were performed in the frequency range from 0.1 to 7 GHz. The initial dimensions of the antennas were optimized in CST microwave studio to achieve low *S*_11_ over 2.45 and 5.18 GHz bands. CST microwave studio is a commercial simulation software used for analysis of electromagnetic structures through solving the Maxwell equations via finite integration technique (FI). An active (radiating) part of the antenna was made of a composite comprising polyurethane filled with multiwall carbon nanotubes (PU/MWCNT).

## 3. Results

### 3.1. Composite Material

Scanning electron microscopy showed homogenous dispergation of multi wall carbon nanotubes in thermoplastic polyurethane as a matrix. The layer was made by simple dipping of the PET foil (the substrate) into the carbon nanotubes dispersion. The uniform layer was created during drying process, by which the solvent was removed. The morphology shows a significant amount of multiwall carbon nanotubes and the polyurethane part around them (Figure 2A). The cross section after cryo-fracture in liquid nitrogen (Figure 2B) shows uniform distribution of carbon nanotubes in vertical and horizontal direction. Another cross-section SEMs (Figure 2C) after cryo-fracture in liquid nitrogen confirms uniform thickness and also double side coating of the substrate by the composite layer. SEM micrograph can be also used for determination of the composite layer thickness (Figure 2D).

Transmission electron microscopy was used to examine morphology of raw carbon nanotubes. The diameter of individual carbon nanotubes is around 25 nm and even the walls are distinguishable (Figure 3A). Carbon nanotubes have typical aggregate into bundles due to Van der Waals forces. To prevent this phenomenon, the sonication technique is used. Thus, macromolecular chains of PU are inserted between the carbon nanotubes to eliminate bundles formation.

The main binding energy peak (284.5 eV) in XPS spectra of MWCNT was assigned to the C1s-sp^2^, while the other ones were assigned to C–O (286.15 eV), C=O (287.1 eV), O–C=O (288.8–289 eV) and C1s-π–π* (291.1–291.5 eV). According to our XPS results of MWCNT total oxygen content was determined to be 18.8 at % for pure MWCNT. The sp^3^/sp^2^ carbon ratios are 2.50 and 1.69 for pure MWCNT (Figure 4).

Thermogravimetric curve presented in Figure 5 shows the weight loss during constant heating rate. Pure multiwall carbon nanotubes (MWCNT) show dramatic weight loss at 632 and 843 °C. Thermoplastic polyurethane (PU) has significant weight drop at 352 °C. Polymer composite (PU/MWCNT) shows two slight weight drops at 313 and 365 °C, respectively, both of which corresponds to polyurethane decomposition; the cumulative drop is around 30 wt % and equals to polyurethane content of the PU/MWCNT composite.

The PU/MWCNT contains 30 wt % of multiwall carbon nanotubes. The height faction of conductive filler dramatically improves electrical conductivity which is crucial point to reach high radiation efficiency and gain of prepared antenna. Measurement of DC conductivity on concentrations of MWCNT is shown in Figure 6. The percolation happens for 20 wt % of MWCNT. For 23 wt % the conductivity around 0.3 S/m was achieved. With increasing concentration of MWCNT above 23%, the conductivity of the PU/MWCNT composite further increases. For concentration 30 wt %, the conductivity 160 S/m was achieved. This point is not shown in the graph for the case of clarity. Right part of the figure shows an impact of conductivity of the composite on radiation of a simple unipole antenna (length 24.9 mm, width 4 mm, 0.5 mm thick Rogers RO4350B substrate). For conductivity 120 S/m, the radiation resistance of antenna at 2.45 GHz around 30 Ohm is obtained. However, for conductivity that is only 12 S/m, the radiation resistance becomes smaller (about 15 Ohms) and the gain of the unipole antenna is decreased. That is the reason why large filling of 30 wt % for our composite was used.

Polymer composite antennas prepared on support substrate is a very novel idea. This fact makes it very difficult to compare with the literature. The inspiration was found on the opposite end of spectra of interest in electromagnetic shielding of GHz waves. There is no exception to use 45 or 50 wt % of conductive filler [32,33].

The highly filled system has some disadvantage which is necessary to solve. The chosen of dip coating technique is one of the easy ways how to prepare very uniform layer without defects. The DFM was chosen as a solvent for PU. PU solution in DMF has very low viscosity which allow to reach relatively high MWCNT content. PU solution also allow to control and adjust viscosity of PU/MWCNT and avoid mechanical degradation of used substrate, no abrasion was observed. SEM analysis also confirm on abrasion on the surface of prepared sample.

### 3.2. Electrical Characterization

Measured DC conductivity of the antenna σ = 160 S/m was compared with AC conductivity extracted from waveguide measurements [30]. Despite the use of the conductive paste between the sample and the waveguide walls, the results were from measurements with the sample inside the waveguide were not acceptable. We believe that this was due to insufficient thickness of our material, which was only 70 μm thick. On the other hand, results when measuring with samples sandwiched between the flanges were better. The DC measurements of conductivity were performed by a simple two-point method since the total resistance of the measured sample was much higher than a contact resistance.

The measured dependence of the relative permittivity (ε_r_) and permeability (μ_r_) on frequency is shown in Figure 7 and Figure 8. The real and imaginary part of permittivity of composites with CNTs is monotonically decreasing with frequency. This would be the case if the composite material was measured inside a rectangular waveguide [30]. However in our case, the measurement took place on the flange of the waveguide, but algorithms being used assumed that samples are placed inside the waveguide. This caused errors in the evaluation, which have two consequences. The first is that the functions ε_r_ and μ_r_ do not decrease too much with frequency within a single waveguide band. The second consequence is that discontinuities arise on the curve when moving from one frequency band to another. Validity of the first and second statements (consequences) were verified by CST microwave studio, where a synthetic composite material having smooth ε_r_ and μ_r_ was analyzed, resulting S-parameters were then transferred to MATLAB (Nattick, USA). Finally, ε_r_ and μ_r_ were extracted by algorithm described in [30]. The results obtained in MATLAB confirmed both previous statements.

The average conductivity at 2.45 GHz is about 290 S/m and average conductivity at 5.18 GHz is around 220 S/m (Figure 8). The average permittivity at 2.45 GHz is 130 and 90 at 5.18 GHz. The behavior of permeability could not be measured too accurately since a very thin (70 μm) sample was used. The average value of the measured permeability was about 2.5 at 2.45 GHz and it decreased with frequency. Since the PU/MWCNT composite material is non-magnetic, the values of permeability should be close to 1.

Nevertheless, after the fabrication of a simple unipole composite based antenna (Rogers substrate RO4350B having thickness *h* = 0.508 mm, unipole length 24.9 mm), it was observed that the agreement between reflection coefficients of simulated and measured antenna was rather poor. The experimentally obtained values of *S*_11_ in dB were approximately 2 dB higher than the value predicted by the full wave simulation program (CST microwave studio). This indicates that the real effective conductivity of the composite is smaller than the value measured on the flange of the waveguide.

Due to low precision of the waveguide methods (for thin samples), the conductivity of our composite at microwave frequencies was finally measured with a microstrip line (see Figure 9). After measurements of the insertion loss *S*_21_ of the sample, the microstrip line was modelled in CST microwave studio (including SMA connectors) and the value of conductivity was changed by optimization in MATLAB to equal simulated and measured *S*_21_ (see Figure 10). The resulting conductivity of our PU/MWCNT material was equal to 120 S/m. Assuming this value of conductivity the typical agreement between measured and simulated *S*_11_ of a simple single layer unipole antenna improved from 2.0 to 0.5 dB.

### 3.3. Design of Antenna

In the case of the design of a grounded unipole antenna printed on a traditional dielectric substrate, the thickness of the antenna (substrate) must be considerable in order to obtain reasonable radiation resistance of the antenna. In the case of PU/MWCNT (or other carbon-based antennas such as buckypaper or PANI/MWCNT), the thickness can be smaller. Thus, low profile antennas can be realized. In Figure 11 a parametric study, performed in CST microwave studio, of a simple unipole made from PU/MWCNT composite is shown. The simulations considered that the unipole is placed on 0.508 mm thick Rogers RO4350B substrate. The length of unipole was 24.9 mm and width 4 mm. It can be seen that with increasing conductivity of the composite radiation resistance increases too; also, the reactance of the antenna starts to oscillate as is typical of antennas made from conductors.

In order to make the antenna operable and to achieve practical radiation efficiency a relatively thick composite layer (t_c_ layer) has to be used. Considering conductivity around 120 S/m for our operating bands, at least 600 μm thick composite has to be used. However, preparation of such a thick composite layer was not feasible from a technological point of view. Therefore, the antenna was made by stacking three layers on top of each other (Figure 12). Assuming the total thickness of the antenna *t*_c_ = 600 μm, the radiation resistance of the antenna of about 30 Ohm was achieved. With further thickness or conductivity increase, radiation resistance upwards of 50 Ohm is achievable. However, the fabrication of an antenna consisting of more than three layers would be technologically problematic.

Matching of the unipole antenna can be done with a LC matching circuit (Figure 13). Either a simple LC match or a double LC can be used. Use of the double LC match can guarantee independent matching in both operating bands. For the purpose of simplicity, the simple LC match was used for the unipole antenna. The length and width of C was 8 and 7 mm respectively. The length and width of L was 2.0 and 1.9 mm respectively.

### 3.4. Fullwave Simulations of the Antenna and Measurements

Based on the procedure described in the previous chapter, a unipole and an F-antenna were designed for dual band operation at 2.45 and 5.18 GHz frequency bands. The design with the F-antenna was inspired by the paper [31]. The difference between our approach and the approach described in [31] is that our design uses a ground plane. The presence of the ground plane is essential for the practical design of antenna for wearable applications.

The carrier substrate of the antennas was 2 mm thick polypropylene. The large thickness was essential in order to secure practical values of antennas’ gain.

The 3D models of the antennas are shown in Figure 14. Both antennas (for dimensions see Table 1) are realized by a sandwiching technique. The dimensions of the substrate (2 mm PP layer) are 50 mm × 70 mm. The antennas were fed by a microstrip port. The reference plane of the port was placed at the beginning of the antenna. The effective permittivity of the PU/MWCNT composite was the same as given in Figure 7. The value of the effective permittivity is smoothly decreasing with frequency. The numerical value of the permittivity was 180 at 1 GHz and 70 at 7 GHz. The effective conductivity was constant and equal to 120 S/m. The permeability of 1 was assumed for the PU/MWCNT material.

The unipole and the F-antenna were manufactured in our laboratory (see Figure 15). The feeding of our antennas was realized by a short 50 Ω section on the PP substrate and the 18 mm long 50 Ω section on the FR4 substrate. The microstrip line and the ground layer of PP material was realized by 35 μm thick copper foil. The foil had a self-adhesive acrylic material. Electrical connection between microstrip lines on PP layer and microstrip line on the FR4 substrate was performed by a conductive silver compound. The same compound was also used for connection of PU/MWCNT composite with the feeding microstrip. The unipole antenna contained also a simple LC matching circuit to improve *S*_11_ of the antenna. The F-antenna does not use the matching since the 50 Ω impedance can be achieved by selection of proper dimensions *l*, *l*_1_ and *l*_2_ of the antenna.

Properties of the manufactured antenna were further examined in detail in our EMC laboratory. Firstly, the *S*_11_ of the antennas was measured and compared with simulations (see Figure 16). For the unipole a very good agreement between measured and simulated *S*_11_ was obtained inside 2.45 GHz operating band. At the 5.18 GHz the measured *S*_11_ resonates at lower frequency than the frequency predicted by simulation. For the F-antenna, correlation between measured and simulated *S*_11_ became worse but the manufactured F-antenna showed an excellent impedance match in both operating bands.

Secondly, radiation properties of both antennas were measured in an anechoic chamber (see Figure 17 and Figure 18). The measured gain at the direction of main lobe is about 2 dBi larger than the gain predicted by CST. This is mainly attributed to the fact, that the model of the antennas in CST did not contained feeding microstrip on FR4 line, impedance matching circuit and the N-connector.

It was observed that both antennas achieved a very good match for both operating bands. The correlation between measured *S*_11_ and simulated *S*_11_ is also quite good.

The agreement of predicted and simulated radiation patterns is not as good as expected. The comparison of measured and simulated gains is given in Table 2. In general, the measured gain was by about 2 dB larger than the simulated values. We mainly attribute this to the fact, that simulation in CST have not assumed feeding by the microstrip on the FR substrate. The model of the antenna in CST does not account for the presence of the N panel connector.

The maximum measured gain of the unipole is −10 dBi and −5.5 dB for 2.45 and 5.18 GHz, respectively. The gain of the F-antenna is smaller since it used a thinner composite layer (*t*_c_ = 70 μm only compared to *t*_c_ = 120 μm for the unipole). Its gain is about −12 dBi at 2.45 GHz and −11.4 dBi at 5.18 GHz. By using thickness *t*_c_ = 120 μm for F-antenna the gain would increase about 2 dBi.

Except of the gain, the radiation efficiency of the designed antennas was also monitored (Table 2). The unipole antenna (*t*_c_ = 600 µm) had radiation efficiency 3.6 and 6.8 percent for both operating bands. The radiation efficiency of the F-antenna was smaller due to a smaller thickness of the F-antenna (*t*_c_ = 420 μm).

## 4. Conclusions

A dual band (2.45 and 5.18 GHz) polymer composite based antenna has been designed and characterized. The antenna is made of a composite comprising polyurethane filled with multiwall carbon nanotubes (PU/MWCNT). Main benefit of used composite material is relative simplicity of its synthesis. Second benefit is flexibility of radiator of prepared antenna.

Although the absolute DC conductivity of the composite is lower (σ = 160 S/m) than for PPy or PEDOT it is still applicable for antenna design. In order to achieve the practical efficiency of radiation and to obtain practical gain the antenna, it is necessary to realize the antennas by means of thick layers.

Using the PU/MWCNT composite, a unipole and dual-band F-antenna on a 2 mm polypropylene (PP) substrate was designed and manufactured.

Measured gain of the unipole antenna is −10.0 and −5.5 dBi for 2.45 and 5.18 GHz frequency bands, The gain of F-antenna is lower than the gain of the unipole since the smaller thickness of the PU/MCNWT composite was used for the F-antenna than the thickness of the composite for the unipole. The gain of the unipole and F-antenna predicted by CST microwave studio did not agreed to much with the measured gains. This is mainly attributed to the fact, that the radiation effect of FR4 microstrip feeding line and the radiation effect of N-connector were not modeled in CST microwave studio.

In addition to the gain, the agreement of the reflection coefficient S_11_ of both antennas obtained by measurement and simulation was also monitored. A good agreement was reached for the unipole. This good agreement was caused by a precise measurement of conductivity of the PU-MWCNT composite at microwave frequencies up to 7 GHz.

The designed antennas are partially bendable and might be practically used in wearable applications. The proposed PU/MWCNT composite material can also find application in other areas than antenna design. For example microstrip matches (terminations), attenuators, as a resistive element in periodic frequency selective surfaces etc.

## Figures and Tables

**Figure 1 polymers-12-02759-f001:**
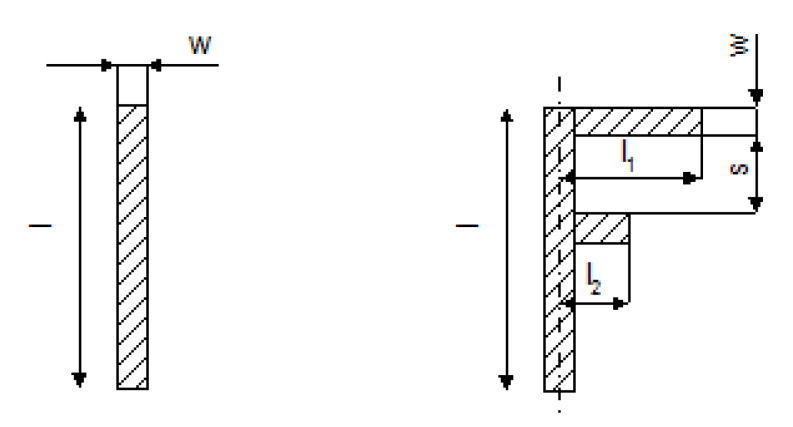
Topology and dimensions of the two studied antenna designs (on the left a dipole; on the right an F-antenna).

**Figure 2 polymers-12-02759-f002:**
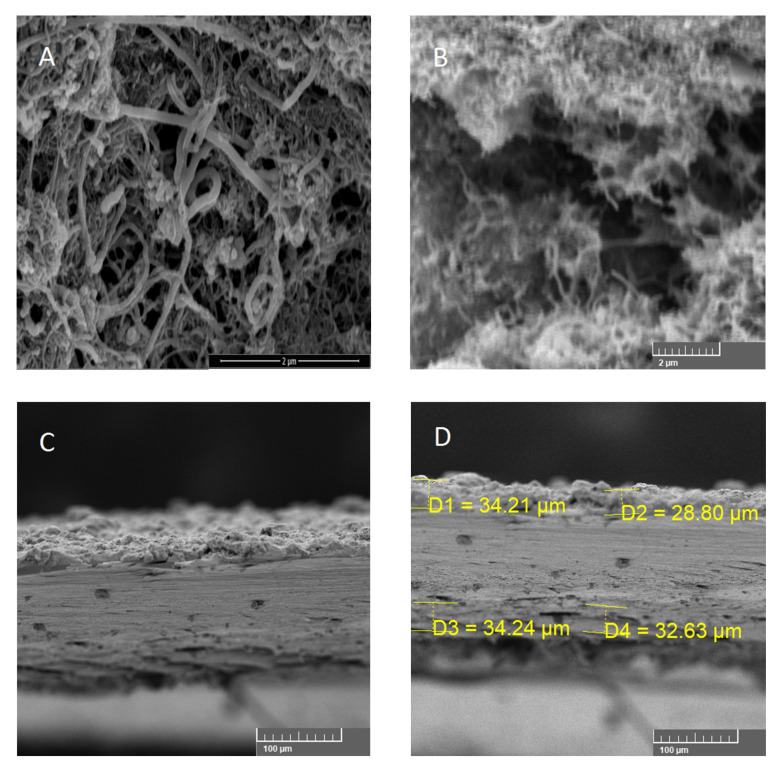
SEM pictures of prepared PU/MWCNT composite. (**A**) surface of PU/MWCNT composite. (**B**) cross section of prepared composite (upper part) (**C**) cross section of PU/MWCNT composite on PET foil. Double side coating. (**D**) Cross section of PU/MWCNT composite with measured thickness (denoted D1–D4).

**Figure 3 polymers-12-02759-f003:**
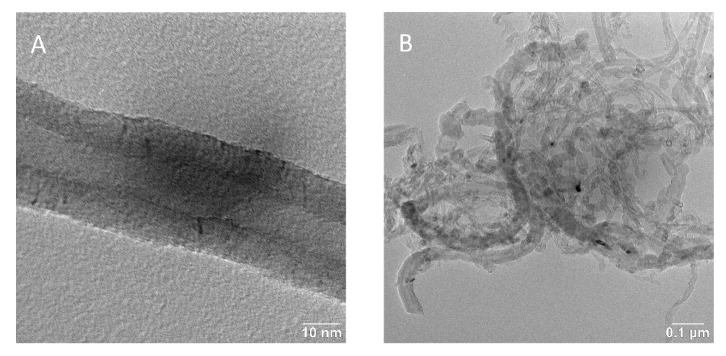
TEM picture of: (**A**) an individual multiwall carbon nanotube, (**B**) a bundle of multiwall carbon nanotubes.

**Figure 4 polymers-12-02759-f004:**
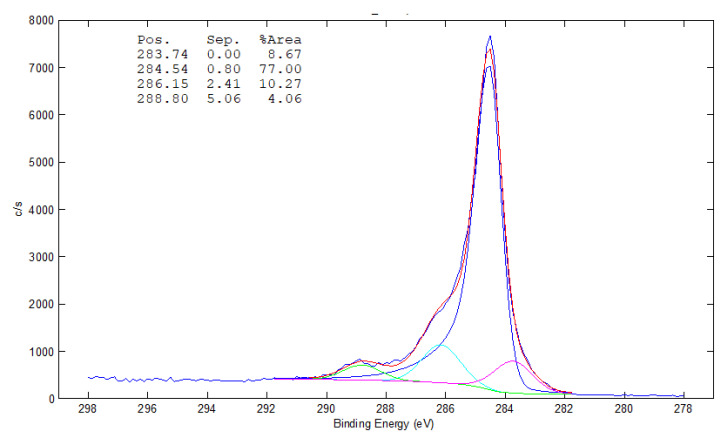
XPS analysis of MWCNT, raw material without any treatment.

**Figure 5 polymers-12-02759-f005:**
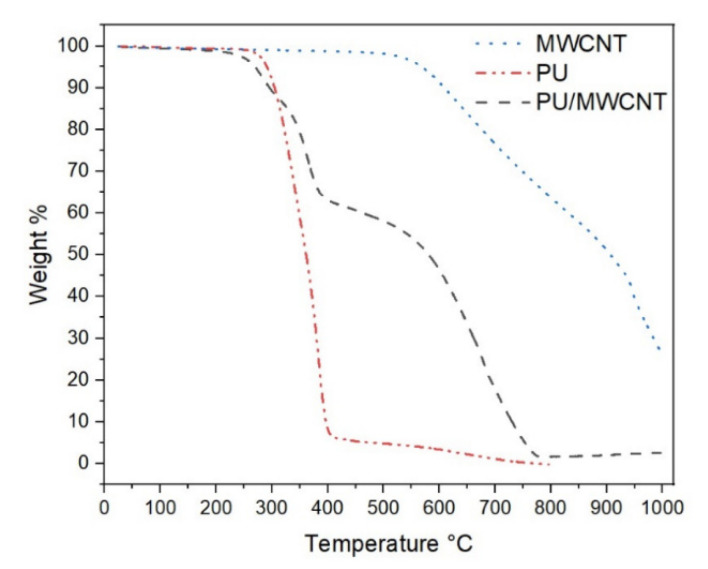
Thermogravimetric (TGA) curves of MWCNT (dot line), Thermoplastic polyurethane PU (dash dot dot line), PU/MWCNT composite (dash line).

**Figure 6 polymers-12-02759-f006:**
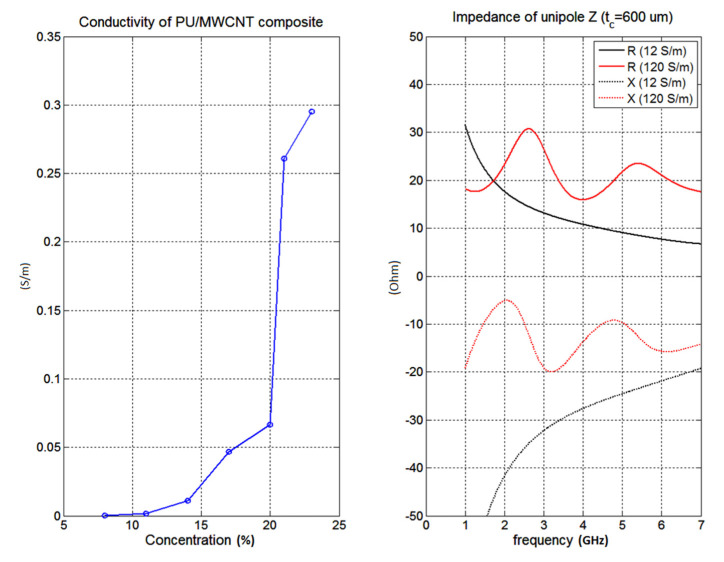
Left (Conductivity of PU/MWCNT composite versus concentration of CNTs), Right (Impedance of unipole antenna).

**Figure 7 polymers-12-02759-f007:**
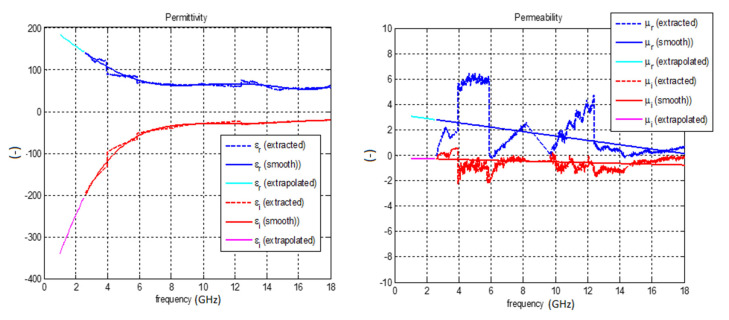
Measured effective permittivity (left) and magnetic permeability (right) of the polymer composite comprising carbon nanotubes and polymer matrix (measurement on the flange of the rectangular waveguide).

**Figure 8 polymers-12-02759-f008:**
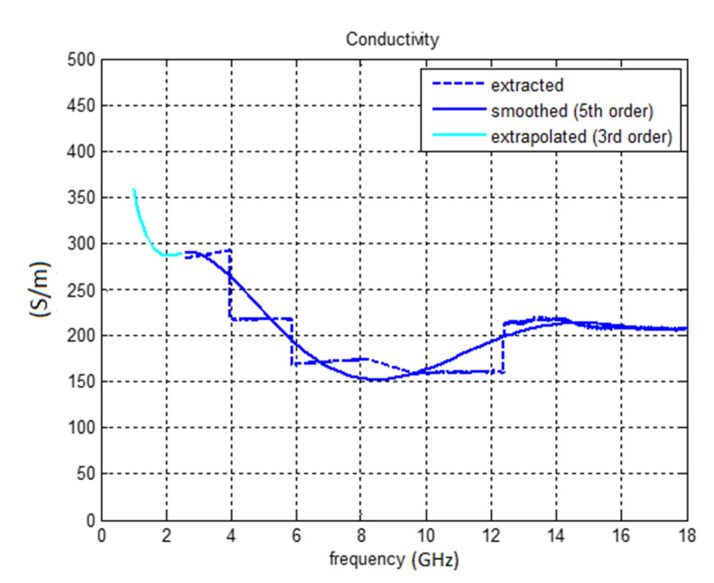
Measured effective conductivity of the polymer composite comprising carbon nanotubes and polymer matrix (measurement on the flange of the rectangular waveguide).

**Figure 9 polymers-12-02759-f009:**
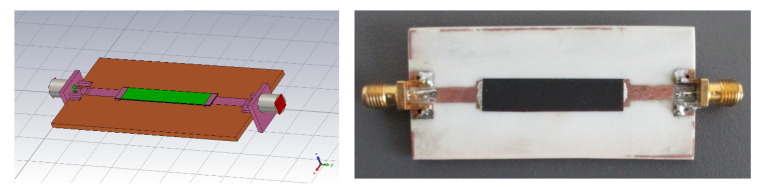
CST studio model of (left) and the real microstrip line (right) made from polymer composite comprising carbon nanotubes and polymer matrix for effective conductivity measurement.

**Figure 10 polymers-12-02759-f010:**
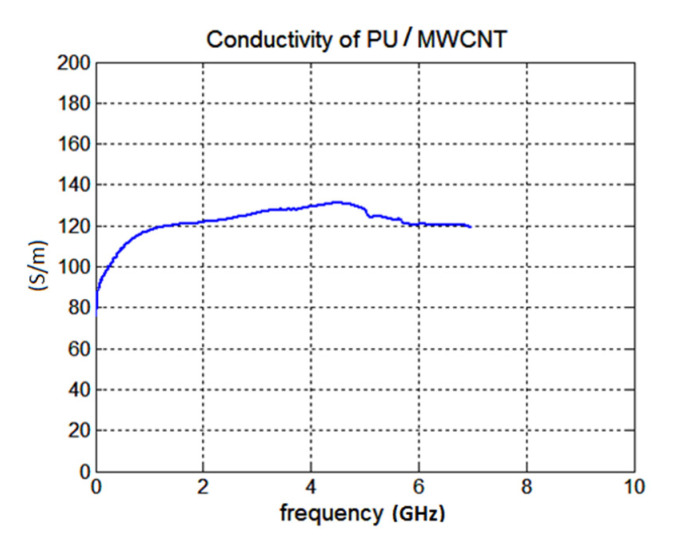
Measured results of the effective conductivity of the PU/MWCNT composite by a microstrip line.

**Figure 11 polymers-12-02759-f011:**
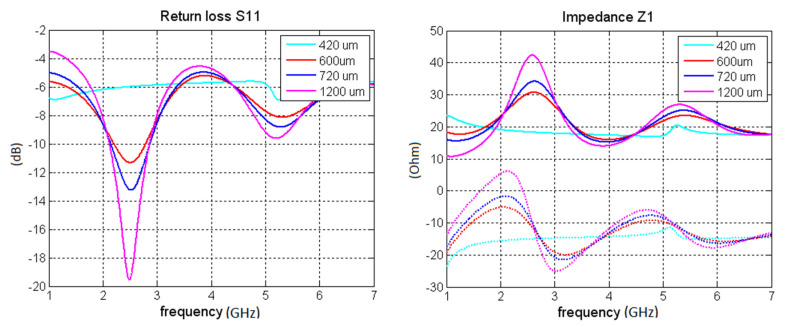
A simulation of characteristics of a unipole made from PU/MWCNT, input impedance as a function of conductivity.

**Figure 12 polymers-12-02759-f012:**
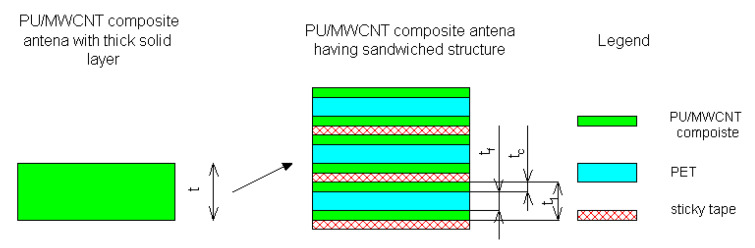
Detailed view of the layered structure of the polymer antenna.

**Figure 13 polymers-12-02759-f013:**
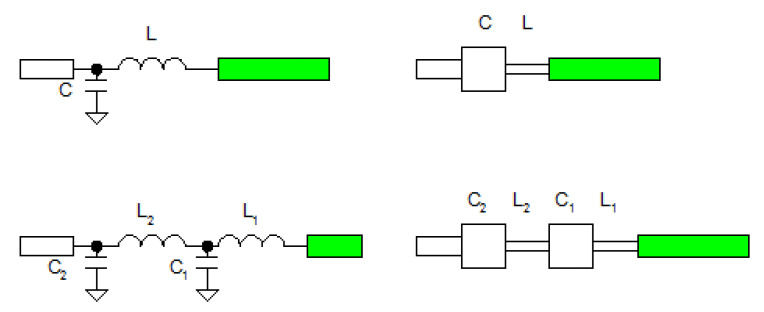
Schematic of the matching circuit of the antenna (top: unipole, simple match, bottom: independent matching in both operating bands).

**Figure 14 polymers-12-02759-f014:**
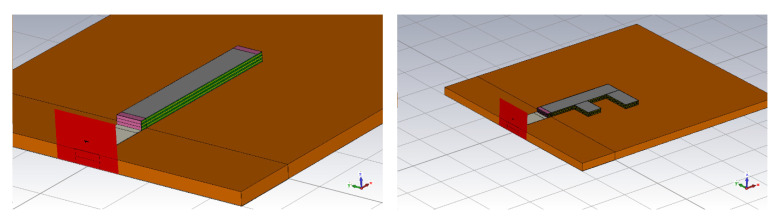
3D model of unipole and F-antenna in CST microwave studio.

**Figure 15 polymers-12-02759-f015:**
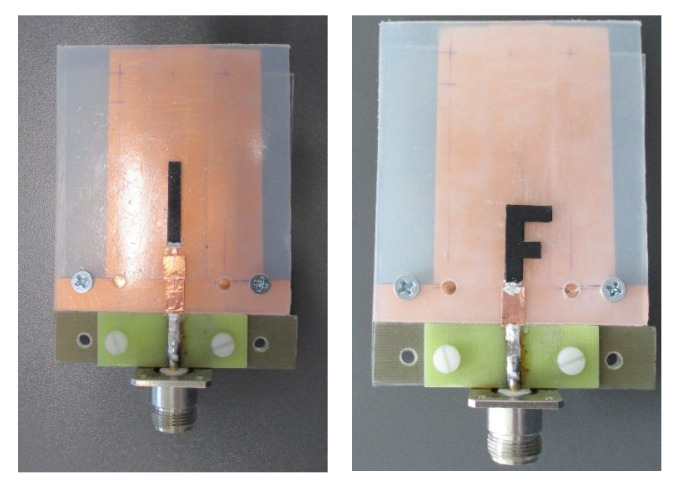
A photo of manufactured antennas (left: unipole; right: F-antenna).

**Figure 16 polymers-12-02759-f016:**
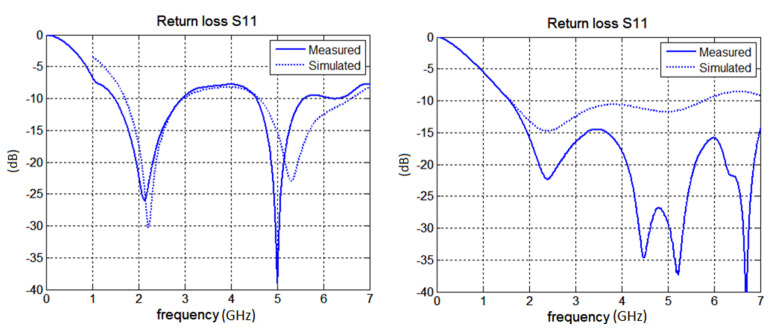
The measured and simualted return loss of antenna (unipole on the left; F-antenna on the right).

**Figure 17 polymers-12-02759-f017:**
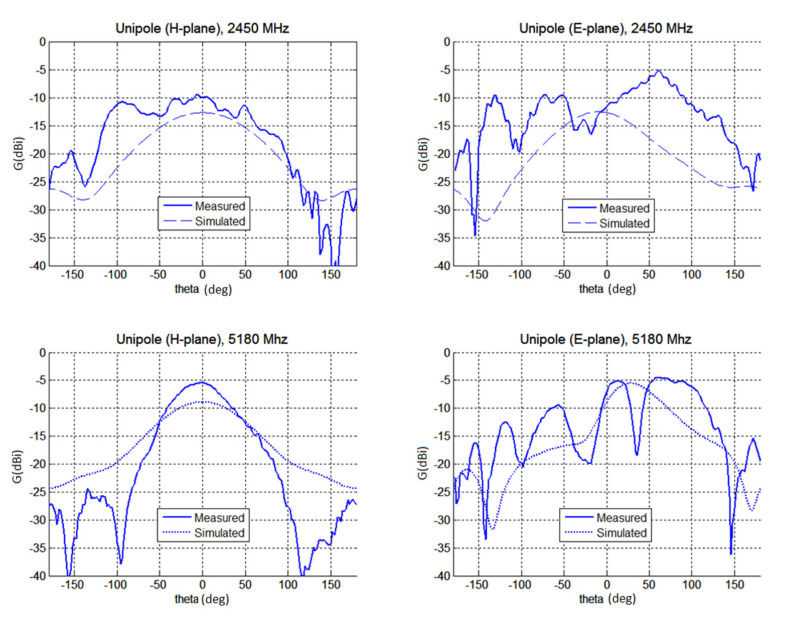
The measured and simualted gain of the unipole antenna (top line, absolute gain for H and E-plane at 2450 MHz, bottom line, absolute gain for H and E-plane for 5180 MHz).

**Figure 18 polymers-12-02759-f018:**
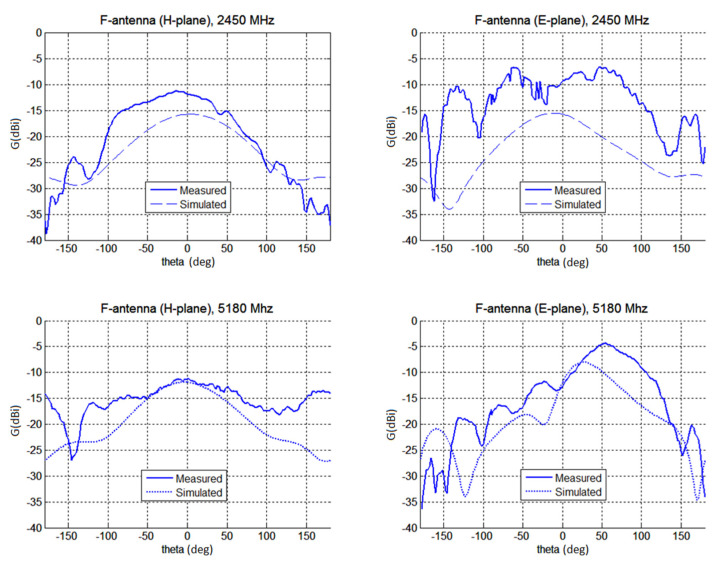
The measured and simulated gain of the F-antenna (top line, absolute gain for H and E-plane at 2450 MHz, bottom line, absolute gain for H and E-plane for 5180 MHz).

**Table 1 polymers-12-02759-t001:** Physical dimensions of the unipole and the F-antenna.

Antenna Type	Parameter	Symbol	Numerical Value
**Unipole**	length of unipole	*l* (mm)	26
	width of unipole	*w* (mm)	4
	thickness of single composite layer	*t*_c_ (μm)	120
	total thickness of unipole (including PET layers)	*t*_tot_ (μm)	1014
**F-antenna**	length of F-antenna	*l* (mm)	21
	length of first arm	*l*_1_ (mm)	11
	length of second arm	*l*_2_ (mm)	7
	width of F-antenna	*w* (mm)	4
	separation between arms	*s* (mm)	5.9
	thickness of single composite layer	*t*_c_ (μm)	70
	total thickness of F-antenna (including PET layers)	*t*_tot_ (μm)	714

**Table 2 polymers-12-02759-t002:** Simulated and measured gain of dipole and F-antenna.

Antenna Type	Frequency (MHz)	Simulated Gain (dBi)	Measured Gain (dBi)	Simulated Radiation Efficiency (%)
Unipole	2450	−12.7	−10.0	3.6
	5180	−8.9	−5.5	6.8
F-antenna	2450	−15.8	−12.0	2.2
	5180	−11.9	−11.4	4.0

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
