# Peer review of "Polyurethane-Carbon Nanotubes Composite Dual Band Antenna for Wearable Applications"

_polymers, 2020, doi:10.3390/polym12112759_

Round 1
Reviewer 1 Report
The manuscript deals with the composite polymer –CNT dual band antennas for practical wearable applications. The work presents a decent combination of experimental and simulation methods.
The paper can be published in “Polymers” after addressing following points.
1) Page 3 line 104
After that 30 % wt. composite was made in the form of dispersion does it mean that the material contains 30% wt. of carbon nanotubes component ? Looks like a very huge fraction. The issue concerning the content of CNTs should be clarified – the wt% of CNT and reasons for using this peculiar value. It is also important to discuss in the materials and methods section how the increase or decrease of the CNT component would affect the overall result of the experiment performed.
2) Page 6 line 182.
On the other hand, results when measuring with samples sandwiched between the flanges were better, physically meaningful and smoothly decreasing with frequency (Fig. 5 and Fig. 6)
Fig. 5 presents the measured effective permittivity (left) and magnetic permeability (right) of the polymer composite 191 comprising carbon nanotubes and polymer matrix. The smoothing approach should be explained for mr (extracted) and mr (smooth)) as well as for mi (extracted) mi (smooth)) since the plot of extracted values with numerous peaks do not look like “smoothly decreasing with frequency”. The nature of registered peaks should be explained as well as the justification for neglecting them during the smoothing procedure.
3) The term “low toxicity of precursors” in the Conclusions section is arguable since the long-range effect of CNT impact is not well investigated.
4) The aspect of the potential practical application should be uncovered in more details
Reviewer 2 Report
In this manuscript, the authors reported the preparation, characterization of Polyurethane-carbon nanotubes composite for application on dual band antenna. This manuscript needs a major revision. The comments are given as below.
- The English language of the manuscript must be polished.
- The references are too few. Following references should be included in the introduction part for more readable, relevant to various conducting polymers. Polymer Engineering & Science, 2019, 59(S1), E33-E43; Polymer Engineering & Science, 2019, 59(S2), E224-E230; Polymers, 2019, 11(3), 546.
- The abbreviation of chemicals should be same, such as thermoplastic polyurethane, multi-wall carbon nanotubes. The abbreviation also should be used in a general method, such as hour.
- The introduction part should be restructured, background, Limitation, novelty and significant contributions of this study should be highlighted in a reasonable sequence.
- The authors used dip technique to fabricate samples, what about the abrasive resistance of the samples?
- The weight ratio of MWCNT in the TPU/MWCNT polymer composites should be offered.
- Line 168,169, What does it (Multiwall carbon nanotubes Sun Nanotech CNT) mean? The authors should confirm the TGA curves of multiwall carbon nanotubes, the weight loss is too high.
- The authors should provide the simulation condition in experiment part.
Round 2
Reviewer 2 Report
The authors solved all the problems.